

# The metabolic profile of a rat model of chronic kidney disease

Yohei Tanada[1,*], Junji Okuda[1,*], Takao Kato[1], Eri Minamino-Muta[1], Ichijiro Murata[2], Tomoyoshi Soga[3], Tetsuo Shioi[1] and Takeshi Kimura[1]

[1] Department of Cardiovascular Medicine, Graduate School of Medicine, Kyoto University, Kyoto, Japan
[2] Department of Chronic Kidney Disease, Gifu University Graduate School of Medicine, Gifu, Japan
[3] Institute for Advanced Biosciences, Keio University, Tsuruoka, Japan
[*] These authors contributed equally to this work.

## ABSTRACT

**Background**. The kidney is always subjected to high metabolic demand. The aim of this study was to characterize metabolic profiles of a rat model of chronic kidney disease (CKD) with cardiorenal syndrome (CRS) induced by prolonged hypertension.

**Methods**. We used inbred male Dahl salt-sensitive (DS) rats fed an 8% NaCl diet from six weeks of age (high-salt; HS group) or a 0.3% NaCl diet as controls (low-salt; LS group). We analyzed function, pathology, metabolome, and the gene expression related to energy metabolism of the kidney.

**Results**. DS rats with a high-salt diet showed hypertension at 11 weeks of age and elevated serum levels of creatinine and blood urea nitrogen with heart failure at 21 weeks of age. The fibrotic area in the kidneys increased at 21 weeks of age. In addition, gene expression related to mitochondrial function was largely decreased. The levels of citrate and isocitrate increased and the gene expression of alpha-ketoglutaratedehydrogenase and succinyl-CoA synthetase decreased; these are enzymes that metabolize citrate and isocitrate, respectively. In addition, the levels of succinate and acetyl Co-A, both of which are metabolites of the tricarboxylic acid (TCA) cycle, decreased.

**Conclusions**. DS rats fed a high-salt diet were deemed a suitable model of CKD with CRS. Gene expression and metabolites related to energy metabolism and mitochondria in the kidney significantly changed in DS rats with hypertension in accordance with the progression of renal injury.

## INTRODUCTION

Chronic kidney disease (CKD) is a worldwide public health problem that affects millions of people worldwide. There has been an increase in the prevalence of CKD in developed and developing countries. The Global Burden of Disease Study revealed that CKD was ranked 27th on the list of causes of total number of global deaths in 1990, but rose to 18th in 2010 (*Lozano et al., 2012*). Patients with CKD have a ten-fold increase in incidence of cardiovascular death due to common underlying causes of cardiovascular diseases (*Hillege et al., 2000*; *Lozano et al., 2012*; *Smith et al., 2006*), and impaired renal function is an independent risk factor for morbidity and mortality in patients with heart failure (HF)

Corresponding author
Takao Kato,
tkato75@kuhp.kyoto-u.ac.jp

(*Hillege et al., 2000*; *Smith et al., 2006*). Therefore, the term cardiorenal syndrome (CRS) has been introduced in recent years to characterize this association.

Many risk factors for kidney disease progression have been identified, and poorly controlled hypertension is a major predictor (*Eirin & Lerman, 2013*; *Nourbakhsh & Singh, 2014*; *Rubattu et al., 2013*). Studies have highlighted several activated deleterious pathways in the hypertensive kidney. Inflammation, oxidative stress, the renin-angiotension-aldosterone system (RAAS), and fibrosis are proposed underling mechanisms eliciting functional decline (*Bidani & Griffin, 2004*; *Eirin & Lerman, 2013*). Moreover, in CRS, the same deleterious pathways develop vicious cycle, which CKD and HF exacerbate each other (*Rubattu et al., 2013*). However, the precise molecular mechanisms responsible for renal injury in CRS have not been fully elucidated. The kidney is a highly energetic organ that is richly perfused. Renal blood flow (RBF) normally accounts for approximately 25% of cardiac output. The renal oxygen consumption per gram of tissue is the second-highest compared to that of other organs, with cardiac oxygen consumption being the highest (*Nourbakhsh & Singh, 2014*). This is largely driven by the high RBF since the renal oxygen extraction is low mitochondrial homeostasis is critical for the maintenance of normal kidney function, including the production of cellular energy and the regulation of reactive oxygen species. Thus, metabolic protection against acute kidney injury has been intensively investigated, such as activators of mitochondrial biogenesis and ROS modulators (*Ishimoto & Inagi, 2016*). However, there are few studies that have explored the influence of metabolic changes of CKD in CRS.

The Dahl salt-sensitive (DS) rat is a useful model system for understanding salt-induced hypertension, since these rats exhibit marked hypertension when they are fed a high-salt diet (*Inoko et al., 1994*; *Kato et al., 2010*). Specifically, when the rats are fed a high-salt diet, they develop hypertension and subsequently develop HF at approximately 18 weeks of age (*Kato et al., 2010*). The DS rats fed a high-salt diet also show an increase of serum creatinine and urinary protein expression related to kidney injury at approximately 10 weeks, and can be used as a model of CKD (*Hisaki et al., 2005*; *Nishiyama et al., 2004*). The purpose of the present study was-to characterize the change of renal metabolic profiles using DS rats with hypertension at 21 weeks of age, which is the end stage of CKD and HF. This was done to gain insight into potential therapeutic targets for CKD in CRS.

## METHODS

### Animals

Animal care and experiments were approved by the Institutional Animal Care and Use Committee of Kyoto University (MedKyo 14183) and conducted following the Guide for Care and Use of Laboratory Animals published by the United States National Institutes of Health. Inbred male DS rats (Japan SLC, Hamamatsu, Shizuoka, Japan) were fed a 0.3% NaCl (low-salt; LS) diet until the age of 6 weeks, after which they were fed an 8% NaCl diet (high-salt; HS) (*Inoko et al., 1994*; *Kato et al., 2010*; *Kawamoto et al., 2015*). The rats in the current study were the same animals used in our previous report (*Tanada et al., 2015*). Therefore, the characterization of hemodynamics in these rats has already been reported.

## Protocols

DS rats fed only an LS diet (LS group) were used as controls, and DS rats fed an HS diet (HS group) were used as a model of renal failure. There were four groups of rats categorized by the amount of salt in the diet (LS or HS) and week of sacrifice (11 weeks or 21 weeks): the 11-week-old LS group ($n = 8$), 11-week-old HS group ($n = 8$), 21-week-old LS group ($n = 12$), and 21-week-old HS group ($n = 11$). Serial measurements of heart rate and blood pressure were performed every two weeks from the age of 11 weeks until sacrifice at 21 weeks.

## Blood and tissue sampling

The rats were anesthetized with inhaled diethyl ether (Wako Pure Chemical Industries, Osaka, Japan) and sacrificed by decapitation without fasting. Blood samples were collected via the right atrium in tubes and centrifuged at 1,500 rpm for 10 min, and blood serum was taken from the supernatant. Measurement of serum creatinine and blood urea nitrogen (BUN) was entrusted to SRL Inc. (Kyoto, Japan). Both kidneys were rapidly removed, snap frozen in liquid nitrogen, and stored at $-80\,°C$ or fixed with 4% paraformaldehyde (PFA).

## Cardiac echocardiography

Transthoracic echocardiography was performed as previously reported (*Kato et al., 2010*). Rats were anesthetized briefly with inhaled diethyl ether (Wako Pure Chemical Industries, Osaka, Japan), and transthoracic echocardiography was performed using a Sonos-5500 echocardiograph (Agilent Technologies, Santa Clara, CA, USA) with a 15-MHz linear transducer. Intraventricular septal thickness (IVSd), left ventricular dimension in the diastolic phase (LVDd), and left ventricular dimension in the systolic phase (LVDs) were measured with M-mode echocardiography, and fractional shortening (FS) was calculated.

## Quantitative reverse transcription-polymerase chain reaction

Total RNA was isolated from the kidneys in each group by the acid guanidinium thiocyanate-phenol-chloroform method. Quantitative reverse transcription-polymerase chain reaction (RT-PCR) was performed as in previous reports (*Inuzuka et al., 2009*; *Niizuma et al., 2012*). The oligonucleotide primers are listed in Table S1. The mRNA level of each gene was standardized using the expression level of the 18S ribosomal RNA as a control.

## Fibrosis of the kidneys

The kidneys were fixed in 4% PFA, embedded in paraffin, and sectioned for histological evaluation. The fibrotic area was quantified in tissue sections with Sirius Red staining, as previously described (*Okuda et al., 2013*).

## Metabolomic analysis of the kidneys

The kidneys were subjected to metabolomic analysis as described previously (*Kato et al., 2010*) using capillary electrophoresis time-of-flight mass spectrometry (CE-TOFMS) (*Soga et al., 2003*). Briefly, frozen kidney tissues were immediately plunged into methanol (0.5 ml) containing internal standards (300 μM each of methionine sulfate for cations and MES for anions) and homogenized for 3 min to inactivate enzymes. 200 μl of deionized water and 500 μl of chloroform were added, and the mixture was thoroughly mixed. The solution

was centrifuged at 15,000 rpm for 15 min at 4 °C and the 600 µl upper aqueous layer was filtered centrifugally through a Milipore-5 kDa cutoff filter to remove proteins. All CE-TOFMS was performed using an Agilent CE capillary electrophoresis system (Agilent Technologies, Santa Clara, CA). We measured the levels of metabolites in glycolysis and the tricarboxylic acid (TCA) cycle, and amino acids and their derivatives.

### Western blotting

Western blotting of kidney tissues was performed as described previously (*Tanada et al., 2015*). The primary antibodies used for Western blotting were as follows: NDUFA9 (1:1,000, Santa Cruz, Dallas, TX, USA) and GAPDH (1:1,000; Cell Signaling, Danvers, MA, USA).

### Statistical analysis

Values are expressed as means $\pm$ the standard error of the mean (SEM). ANOVA was used for comparisons multiple groups. In all tests, a value of $p < 0.05$ was considered statistically significant.

## RESULTS

### Dahl salt-sensitive rats fed a high-salt diet show a high heart rate, systolic blood pressure, and cardiac dysfunction

As we previously reported (*Tanada et al., 2015*), DS rats fed an HS diet (HS group) developed hypertension, showed cardiac dysfunction, and died from heart failure. DS rats fed an LS diet (LS group) did not develop hypertension, and were used as controls. The heart rate was significantly higher in the HS group compared to the LS group from 11 weeks of age to 19 weeks of age (Fig. S1A). Conversely, systolic blood pressure was significantly higher in the HS group compared to LS group from 11 weeks of age to 21 weeks of age (Fig. S1A). At 21 weeks of age, intraventricular septal thickness (IVSd) increased in the HS group on echocardiographic examination. In addition, left ventricular dimension in the diastolic phase (LVDd) increased, and fractional shortening (FS) decreased in the HS group compared to that of the LS group (Fig. S1B).

### The HS group had a high concentration of serum creatinine and blood urea nitrogen at 21 weeks of age

There was no significant difference in the concentration of serum creatinine or BUN between the LS group and HS group at 11 weeks of age (Fig. S1B). However, the HS group showed a significantly higher concentration of both creatinine and BUN compared to the LS group at 21 weeks of age (Fig. S1C).

### Fibrosis in the kidney and gene expression related to the glomerulus, renal tubules, and fibrosis

At 21 weeks of age, the area of fibrosis in the kidney was significantly increased in the HS group compared to the LS group. There was no difference between the HS and LS groups at 11 weeks of age (Figs. S2A and S2B).

Nephrin and podocin are extracellular components of the slit diaphragm coded by *Nphs1* and *Nphs2*, respectively (*Boute et al., 2000*; *Ruotsalainen et al., 1999*). The expression of *Nphs1* was significantly decreased in the HS group compared to the LS group at 21 weeks

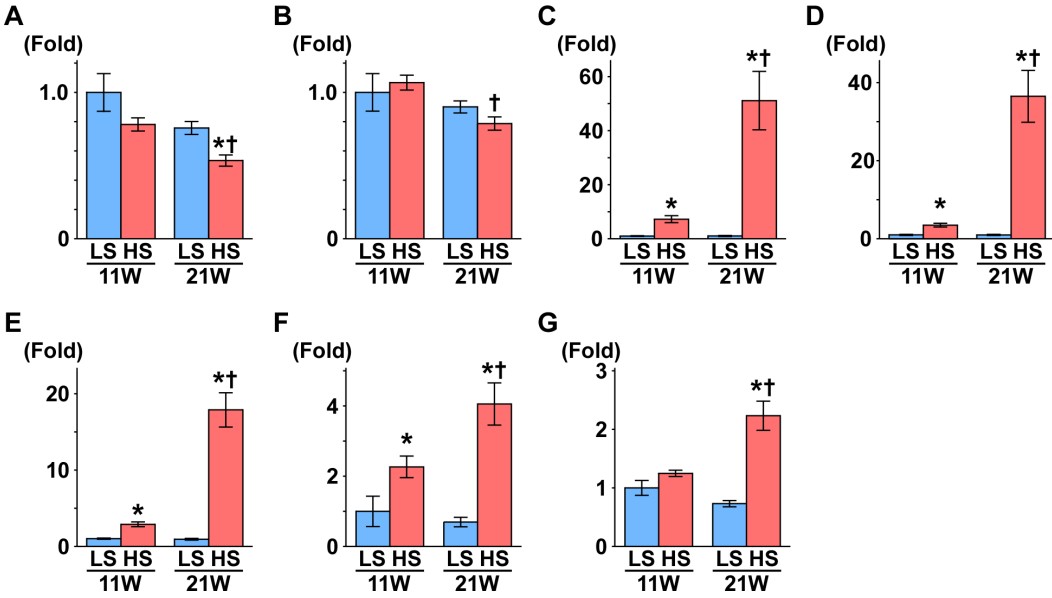

**Figure 1  Gene expression related to the glomerulus, renal tube, and fibrosis.** (A) NPHS1. (B) NPHS2. (C) KIM1. (D) NGAL. (E) Osteopontin. (F) Collagen 1. (G) αSMA. At 11 weeks of age, the expressions of Kim1, Ngal, osteopontin, and collagen 1 were significantly increased in the HS group compared to the LS group. At 21 weeks of age, the expression of Nphs1 significantly decreased in the HS group compared to the LS group. Conversely, the expressions of Nphs2, Kim1, Ngal, osteopontin, collagen 1, and αSMA significantly increased in the HS group compared to the LS group. LS-11 week, $n = 8$; LS-21 week, $n = 12$; HS-11 week, $n = 8$; HS-21 week, $n = 11$. *$p < 0.05$ versus LS-C group; †$p < 0.05$ versus rats at 11 weeks of age.

of age (Fig. 1A). Though the expression of *Nphs2* decreased in the HS group at 21 weeks of age compared to 11 weeks of age, there was no difference between the HS and LS groups at 21 weeks of age (Fig. 1B).

The gene expression of kidney injury molecule-1 (*Kim1*) indicates the degree of interstitial fibrosis, renal tubular damage, and inflammation (*Van Timmeren et al., 2006*). Neutrophil gelatinase–associated lipocalin (NGAL) is also associated with renal tubular damage in ischemic or inflammatory responses (*Mori et al., 2005*). In the present study, the expression of *Kim1* and *Ngal* was significantly higher in the HS group than the LS group at both 11 and 21 weeks of age (Figs. 1C and 1D). In addition, in the HS group, the expression of these genes was significantly increased at 21 weeks of age compared to 11 weeks of age (Figs. 1C and 1D).

The expression of the osteopontin gene increases in rat renal fibroblasts with aldosterone (*Irita et al., 2008*) and in the kidneys of hypertensive rats (*Blasi et al., 2003*). Similarly, the expression of type 1 collagen (collagen 1) and α-smooth muscle actin genes (αSMA) increases in renal fibrosis (*Lekgabe et al., 2005*; *Wang et al., 2008*). In the present study, expression of these genes was significantly higher in the HS group than the LS group at both 11 and 21 weeks of age (Figs. 1E–1G). The expression of these genes increased significantly at 21 weeks of age compared to 11 weeks in the HS group only (Figs. 1E–1G).

## Gene expression related to mitochondrial function, glycolysis, and free fatty acid metabolism in the kidneys

We examined the expression level of some genes related to mitochondrial function, glycolysis, and fatty acid metabolism at 11 and 21 weeks of age. Gene expression related to glycolysis was measured, and the expression of *hexokinase (Hk) 1* and *Hk2*, which catalyze the transfer from glucose to glucose-6-phosphate (G6P), significantly increased in the HS group compared to the LS group at 21 weeks of age. Conversely, the expression of Tp53-induced glycolysis and apoptosis regulator (*TIGAR*), which alters the concentration of fructose 2,6-bisphosphate (*Bensaad et al., 2006*), was significantly decreased in the HS group compared to the LS group at 21 weeks of age (Table 1).

Peroxisome proliferator-activated receptor $\gamma$ coactivator1-$\alpha$ (PGC1-$\alpha$) is a master regulator of mitochondrial biogenesis, though there was no change between the LS and HS group (Table 1 and Fig. 2A). Proliferator activated receptor $\alpha$ (PPAR$\alpha$), nuclear respiratory factor 1 (NRF1), and NRF2 are related to mitochondrial biogenesis. *Ppar$\alpha$* expression was significantly decreased and *Nrf1* expression was significantly increased in the HS group compared to the LS group at 21 weeks of age. Otherwise, compared to that in the LS group, *Nrf2* expression was significantly increased at 11 weeks of age and decreased at 21 weeks of age in the HS group (Table 1 and Fig. 2).

Subsequently, we measured the expression of genes related to oxidative phosphorylation. Though uncoupling protein 3 (UCP3) is known for uncoupling ATP synthesis from oxidative metabolism, its transcription is regulated by PPAR$\alpha$ (*Young et al., 2001*). In this study, like *Ppar$\alpha$* expression, *Ucp3* expression decreased significantly in the HS group compared to the LS group at 21 weeks of age. The expression levels of several genes related to the respiratory chain also changed in the HS group; NADH dehydrogenase 4 (*Nd4*), alpha-subcomplex 9 ($\alpha$S9), succinate dehydrogenase b (*Sdhb*) and cytochrome-c (*Cyt-c*) expression significantly decreased in the HS group from that in the LS group at 21 weeks of age. Conversely, in the HS group, expression of cytochrome c oxidase 1 (*Cox1*) decreased at 11 weeks of age and increased at 21 weeks of age, and *Cox4* increased significantly at 21 weeks of age (Table 1 and Fig. 2).

Medium-chain acyl-CoA dehydrogenase (MCAD) is an enzyme related to fatty acid oxidation, and its expression decreased significantly in the HS group compared to the LS group at 21 weeks of age. ATP-binding cassette sub-family A member 1 (ABCA1) has a critical role in cellular cholesterol and phospholipid efflux, and its gene expression was significantly increased in the HS group compared to the LS group at 21 weeks of age (Table 1).

## Metabolomic profile of the kidney

To examine energy metabolism in renal damage, we performed a comprehensive metabolomic analysis. The results of the quantification showed metabolites related to glycolysis, including the TCA cycle, and amino acids and their derivatives (Fig. 3 and Table S3). By two-way ANOVA analysis, we assessed the significance ($p$ value) for time (11 weeks and 21 weeks) and treatment effects (LS diet and HS diet). The metabolites that were not significantly different ($p \geq 0.05$) regarding time or interaction effects are only represented

**Table 1 Gene expression quantified by real-time RT-PCR in kidney.**

| | 11 week | | High-salt | |
|---|---|---|---|---|
| | **Low-salt** | **High-salt** | **Low-salt** | **High-salt** |
| Numbers of animals | 8 | 12 | 8 | 11 |
| **Glycolysis** | | | | |
| Glut1 | $1.00 \pm 0.08$ | $1.07 \pm 0.07$ | $0.90 \pm 0.07$ | $1.22 \pm 0.12$ |
| Glut4 | $1.00 \pm 0.15$ | $2.43 \pm 0.43$* | $2.41 \pm 0.65$ | $2.64 \pm 0.49$ |
| HK1 | $1.00 \pm 0.10$ | $1.11 \pm 0.08$ | $0.83 \pm 0.04$ | $1.93 \pm 0.23$** |
| HK2 | $1.00 \pm 0.21$ | $1.14 \pm 0.07$ | $0.99 \pm 0.12$ | $2.11 \pm 0.18$** |
| PFK | $1.00 \pm 0.11$ | $1.00 \pm 0.07$ | $0.83 \pm 0.05$ | $0.84 \pm 0.06$ |
| GAPDH | $1.00 \pm 0.24$ | $0.97 \pm 0.10$ | $1.15 \pm 0.13$ | $1.05 \pm 0.17$ |
| PGAM2 | $1.00 \pm 0.27$ | $1.02 \pm 0.10$ | $1.07 \pm 0.11$ | $0.91 \pm 0.13$ |
| TIGAR | $1.00 \pm 0.06$ | $1.01 \pm 0.37$ | $0.98 \pm 0.05$ | $0.69 \pm 0.04$** |
| HIF-1$\alpha$ | $1.00 \pm 0.11$ | $1.01 \pm 0.04$ | $1.00 \pm 0.10$ | $1.21 \pm 0.11$ |
| ACLY | $1.00 \pm 0.11$ | $0.98 \pm 0.05$ | $0.84 \pm 0.06$ | $0.94 \pm 0.08$ |
| **Fatty acids** | | | | |
| ACS | $1.00 \pm 0.15$ | $0.96 \pm 0.10$ | $0.78 \pm 0.06$ | $0.79 \pm 0.09$ |
| CPT-1 | $1.00 \pm 0.33$ | $0.64 \pm 0.11$ | $0.62 \pm 0.17$ | $0.44 \pm 0.10$ |
| ACC$\alpha$ | $1.00 \pm 0.15$ | $1.01 \pm 0.07$ | $0.90 \pm 0.09$ | $1.21 \pm 0.14$ |
| ACC$\beta$ | $1.00 \pm 0.09$ | $0.90 \pm 0.08$ | $1.06 \pm 0.10$ | $0.90 \pm 0.11$ |
| FAT/CD36 | $1.00 \pm 0.14$ | $1.19 \pm 0.07$ | $1.03 \pm 0.09$ | $1.42 \pm 0.15$ |
| VLCAD | $1.00 \pm 0.23$ | $0.81 \pm 0.09$ | $0.82 \pm 0.13$ | $0.58 \pm 0.07$ |
| LCAD | $1.00 \pm 0.29$ | $0.89 \pm 0.11$ | $1.01 \pm 0.14$ | $0.84 \pm 0.15$ |
| MCAD | $1.00 \pm 0.06$ | $1.09 \pm 0.04$ | $1.02 \pm 0.06$ | $0.66 \pm 0.58$** |
| ABCA1 | $1.00 \pm 0.12$ | $1.11 \pm 0.07$ | $0.89 \pm 0.03$ | $1.65 \pm 0.13$** |
| SREBF1 | $1.00 \pm 0.17$ | $0.78 \pm 0.05$ | $0.92 \pm 0.11$ | $0.73 \pm 0.08$ |
| SREBF2 | $1.00 \pm 0.26$ | $0.83 \pm 0.09$ | $0.96 \pm 0.14$ | $0.79 \pm 0.11$ |
| **Mitochondrial function** | | | | |
| PGC1$\alpha$ | $1.00 \pm 0.10$ | $0.85 \pm 0.03$ | $0.95 \pm 0.06$ | $1.04 \pm 0.07$ |
| PPAR$\alpha$ | $1.00 \pm 0.12$ | $0.90 \pm 0.06$ | $1.11 \pm 0.08$ | $0.78 \pm 0.09$** |
| NRF1 | $1.00 \pm 0.10$ | $0.85 \pm 0.02$ | $0.86 \pm 0.04$ | $1.18 \pm 0.07$** |
| NRF2 | $1.00 \pm 0.08$ | $0.71 \pm 0.05$* | $0.72 \pm 0.03$ | $0.84 \pm 0.04$** |
| TFAM | $1.00 \pm 0.04$ | $1.03 \pm 0.03$ | $0.98 \pm 0.08$ | $1.00 \pm 0.05$ |
| UCP3 | $1.00 \pm 0.04$ | $1.13 \pm 0.04$* | $1.19 \pm 0.08$ | $0.89 \pm 0.03$** |
| ANT | $1.00 \pm 0.17$ | $1.00 \pm 0.18$ | $1.12 \pm 0.23$ | $0.94 \pm 0.16$ |
| SDHB | $1.00 \pm 0.05$ | $1.10 \pm 0.05$ | $0.97 \pm 0.07$ | $0.79 \pm 0.04$** |
| $\alpha$S9 | $1.00 \pm 0.03$ | $0.97 \pm 0.02$ | $0.93 \pm 0.02$ | $0.70 \pm 0.04$** |
| ND4 | $1.00 \pm 0.10$ | $0.80 \pm 0.02$* | $0.92 \pm 0.03$ | $0.65 \pm 0.04$** |
| Cyt-b | $1.00 \pm 0.24$ | $0.77 \pm 0.12$ | $0.55 \pm 0.19$ | $0.61 \pm 0.17$ |
| Cyt-c | $1.00 \pm 0.28$ | $1.09 \pm 0.15$ | $1.06 \pm 0.20$ | $0.78 \pm 0.27$** |
| COX1 | $1.00 \pm 0.12$ | $0.85 \pm 0.04$ | $0.92 \pm 0.05$ | $0.68 \pm 0.07$** |

**Table 1** (*continued*)

| | 11 week | | High-salt | |
| --- | --- | --- | --- | --- |
| | **Low-salt** | **High-salt** | **Low-salt** | **High-salt** |
| COX4 | $1.00 \pm 0.07$ | $1.01 \pm 0.03$ | $1.01 \pm 0.05$ | $0.80 \pm 0.06^{**}$ |
| COX5a | $1.00 \pm 0.15$ | $1.08 \pm 0.10$ | $1.31 \pm 0.18$ | $1.08 \pm 0.19$ |

**Notes.**

Glut, Glucose transporter; HK, hexokinase; PFK, phosphofructokinase; GAPDH, glyceraldehyde-3-phosphate dehydrogenase; PGAM, phosphoglycerate mutase; TIGER, TP53 induced glycolysis regulatory phosphatase; HIF-1$\alpha$, Hypoxia Inducible Factor 1-$\alpha$; ACLY, ATP citrate lyase; ACS, Acyl-CoA synthetase; CPT-1, Carnitine palmitoyltransferase-1; ACC, Acetyl-CoA Carboxylase; FAT/CD36, fatty acid translocase/cluster of differentiation 36; VLCAD, very-long-chain acyl-CoA dehydrogenase; LCAD, long-chain acyl-CoA dehydrogenase; MCAD, medium-chain acyl-CoA dehydrogenase; ABCA1, ATP-Binding Cassette 1; SREBF, sterol regulatory element binding transcription factor; PGC1-$\alpha$, peroxisome proliferator-activated receptor $\gamma$ coactivator1-$\alpha$; PPAR$\alpha$, peroxisome proliferator-activated receptor $\alpha$; NRF, nuclear respiratory factor; TFAM, mitochondrial transcription factor A; UCP3, uncoupling protein 3; ANT, adenine nucleotide translocator; SDHB, succinate dehydrogenase b; $\alpha$S9, alpha-subcomplex 9; ND4, NADH dehydrogenase 4; Cyt, cytochrome; COX, cytochrome c oxidase.

Values are the mean $\pm$ SEM.

$^*p < 0.05$ versus Low-salt 11 week.

$^{**}p < 0.05$ versus Low-Salt 21 week.

Red indicates the increase; blue indicates the decrease.

by the bar plots of 21-week-old rats. Other metabolites are represented by bar plots of 11-week-old and 21-week-old rats (Fig. 3). The metabolites of kidney tissue significantly changed in the HS group from that in the LS group. Especially, the increased urea and creatinine levels at 21 weeks of age validated that the metabolome analysis was performed successfully.

The levels of 2,3-diphosphoglycerate (2,3-DPG) and phosphoenolpyruvate (PEP), intermediate metabolites of glycolysis, decreased in the HS group at 21 weeks of age. The levels of several amino acids and derivatives such as serine, betaine, alanine, histidine, citruline, proline, arginine and putrescine, increased in the HS group at 21 weeks of age. On the contrary, taurine, $\beta$-alanine, and glutamate were decreased in the HS group at 21 weeks of age. Levels of several metabolites in the TCA cycle also changed; the levels of citrate, cis-acotinate and isocitrate increased, and the levels of succinate and acetyl CoA decreased (Fig. 3 and Table S3).

Subsequently, we measured the gene expression of enzymes that are involved in the TCA cycle (Fig. 4): aconitase is encoded by *Aco2,* which catalyzes the reversible isomerization of citrate and isocitrate; 2-oxoglutarate dehydrogenase (OGDH) is an enzyme that catalyzes the oxidation of 2-oxoglutarate; succinyl-CoA synthetase (SCS) is encoded by the gene *Suclg1,* and is an enzyme that hydrolyzes succinyl-CoA into succinate and CoA-SH; fumarate hydratase is encoded by the *Fh* gene and is the enzyme that catalyzes the hydration of fumarate to malate. The gene expression of *Aco2*, *Ogdh*, *Suclg1*, and *Fh* was significantly decreased in the HS group compared to the LS group at 21 weeks of age (Fig. 4).

Furthermore, the amounts of nicotinamide adenine dinucleotide (NAD+) and NADH were measured on metabolome analysis. NAD+ plays a central role in cellular energy metabolism and energy production. The NAD+/NADH ratio is often considered as the key of mitochondrial function. Cytoplasmic NAD+/NADH ratios range between 60 and 700, and mitochondrial NAD+/NADH ratios are maintained at 7–8 (*Stein & Imai, 2012*). The reduction of NAD+/NADH ratio has been reported in renal injuries in a type 1 diabetic mouse model (*Zhu et al., 2014*). In the present study, both NAD+ and NADH

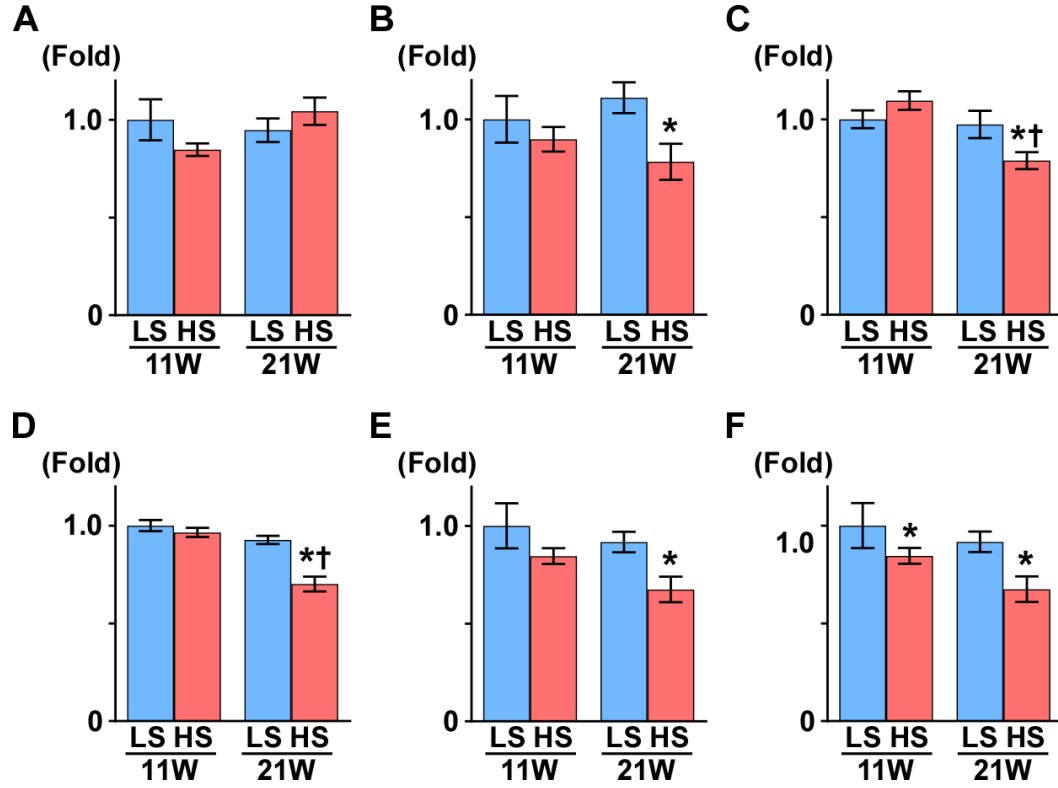

**Figure 2** **Gene expression related to mitochondrial function.** (A) PGC1-$\alpha$. (B) PPAR$\alpha$. (C) SDHB. (D) $\alpha$S9. (E) COX1. (F) COX4. At 11 weeks of age, the expression of COX4 was significantly decreased in the HS group compared to the LS group. At 21 weeks of age, the expressions of PPAR$\alpha$, SDHB, $\alpha$S9, COX1, and COX4 significantly decreased in the HS group compared to the LS group. LS-11week, $n = 8$; LS-21week, $n = 12$; HS-11week, $n = 8$; HS-21week, $n = 11$. *$p < 0.05$ versus LS-C group; †$p < 0.05$ versus rats at 11 weeks of age.

levels significantly decreased at 21 weeks of age, while NAD+/NADH ratio showed no change between the HS group and LS group (Fig. S3B). It is well known that mitochondrial Complex I catalyzes the first step of NADH oxidation, and increases the NAD+/NADH ratio. Therefore, we measured the protein expression of NADH dehydrogenase ubiquinone 1 $\alpha$ subcomplex subunit 9 (NDUFA9), a subunit of mitochondrial Complex I, at 21 weeks of age, because the level of mRNA expression of NDUFA9 ($\alpha$S9) was significantly decreased in the HS group (Table 1 and Fig. 2D). However, there was no significant change between the LS group and HS group (Fig. S3B).

## DISCUSSION

Consistent with previous reports (*Hisaki et al., 2005*; *Karlsen et al., 1997*; *Kato et al., 2010*; *Rapp & Dene, 1985*), DS rats fed an HS diet showed high blood pressure, an increased concentration of serum creatinine and BUN, and an increased fibrotic zone in their kidneys at 21 weeks of age. We demonstrated that the metabolites levels in the first half of the TCA cycle increased; the level of succinate in the latter half of the TCA cycle decreased

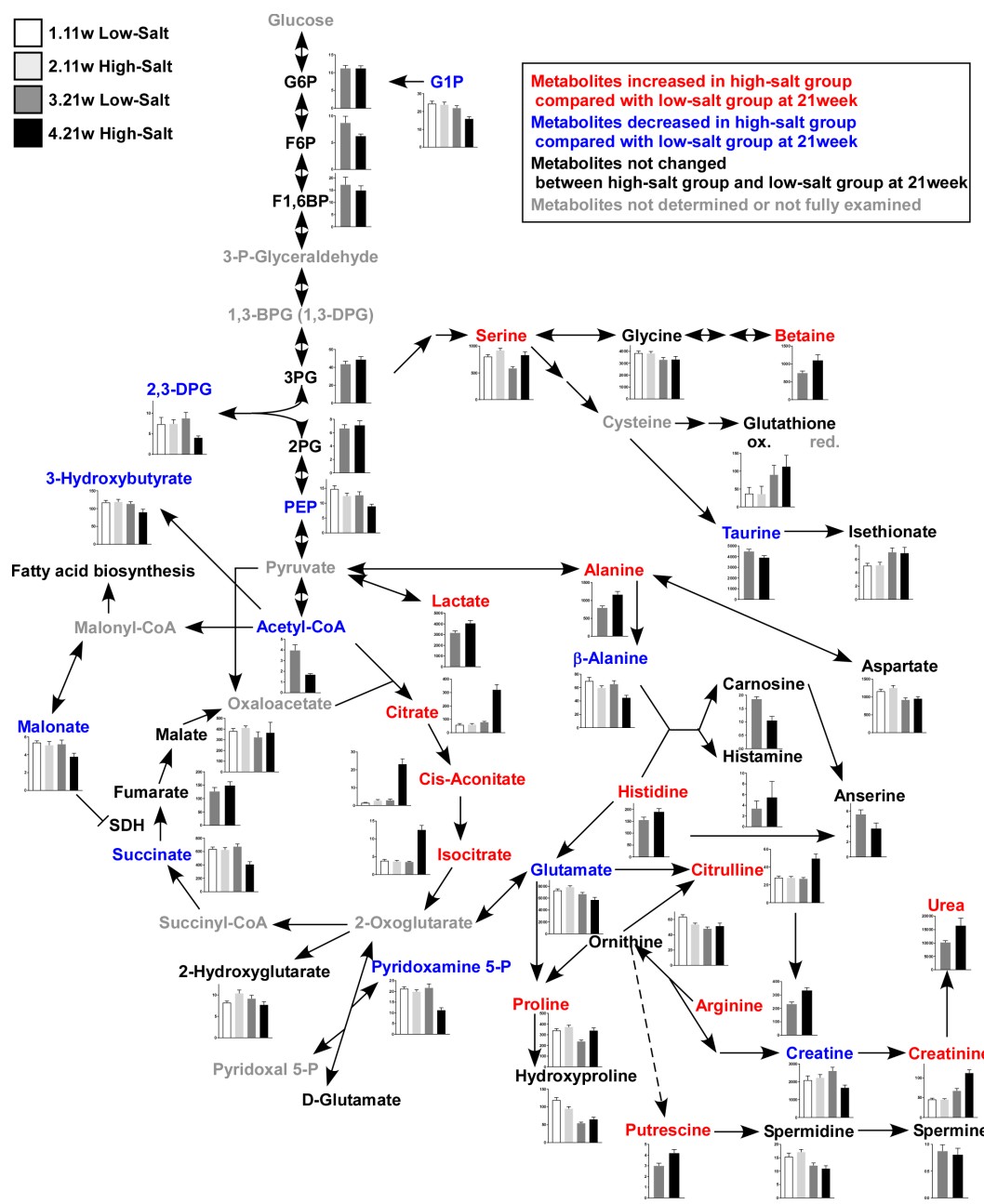

**Figure 3  Metabolites in kidney tissue.** After assessing the *p* value for the time effect (11 weeks and 21 weeks) and the treatment effect (LS diet and HS diet), the metabolites that were not significantly different ($p \geq 0.05$) regarding time or interaction effect are only represent by bar plots of 21-week-old rats. Other metabolites are represented by bar plots of 11-week-old and 21-weeks-old rats. In the HS group at 21 weeks of age, the levels of 2,3-diphosphoglycerate (2,3-DPG) and phosphoenolpyruvate (PEP), intermediate metabolites of glycolysis, decreased. The levels of several amino acids and their derivatives also changed in rats in the HS group at 21 weeks of age. The amount of urea, creatinine, and citrulline increased at 21 weeks of age. The levels of several metabolites in the TCA cycle also changed; the levels of citrate and isocitrate increased while that of succinate and acetyl CoA decreased. LS-11 week, $n = 8$; LS-11 week, $n = 12$; HS-11 week, $n = 8$; HS-21 week, $n = 11$.

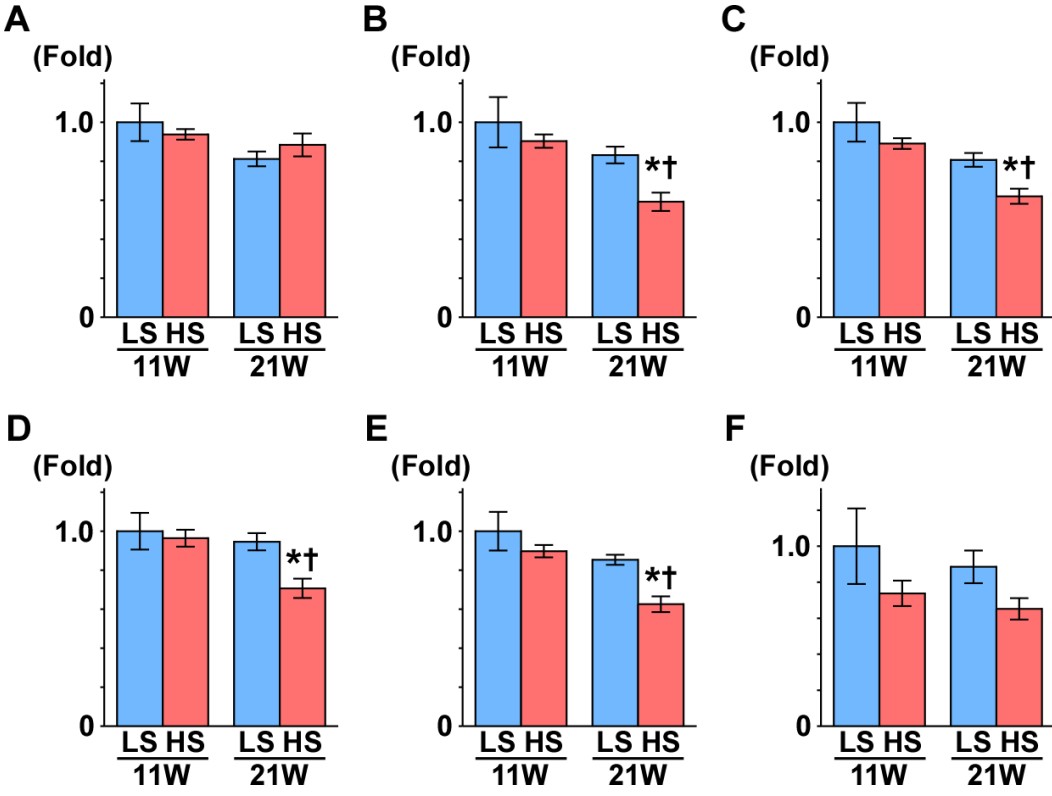

**Figure 4** **Gene expression of enzymes in the TCA cycle.** (A) CS. (B) Aco2. (C) OGDH. (D) Sulg1.(E) Fh. (F) MDH2. The expressions of Aco2, Ogdh, Suclg1, and Fh significantly decreased in the HS group at 21 weeks of age. These results indicate mitochondrial dysfunction related to the TCA cycle. LS-11week, $n = 8$; LS-21week, $n = 12$; HS-11week, $n = 8$; HS-21week, $n = 11$ *$p < 0.05$ versus LS-C group; †$p < 0.05$ versus rats at 11 weeks of age.

in DS rats in CKD with CRS, and the gene expression related to energy metabolism also significantly changed.

The DS rats fed a high-salt diet showed an increase in urinary protein expression and serum creatinine levels, and a decrease in creatinine clearance. They also exhibited injured glomeruli characterized by sclerosis enlarged glomerular size, and severe tubulointerstitial fibrosis on histochemical analysis (*Du et al., 2009*). Moreover, DS rats treated with vasopressin type 2 receptor antagonist showed an improvement in cardiac dysfunction and kidney dysfunction simultaneously (*Ishikawa et al., 2013*; *Morooka et al., 2012*). Treatment with the calcium sensitizer and the AT1-receptor antagonist in DS rats also ameliorate cardiac dysfunction and kidney injury simultaneously (*Biala et al., 2011*). We showed a close correlation among systolic blood pressure, cardiac dysfunction, and renal damage. These findings suggested that the DS rat is a useful model of CKD with CRS. Although this model did not fully recapitulate human hypertensive renal disease, the Dahl rat model nevertheless may have many advantages in preclinical studies for determination of suitable candidates for biomarker and drug discovery.

In the present study, the metabolic profiles in kidney tissues revealed increased levels of citrate and isocitrate, decreased gene expression of *Ogdh* and *Scs,* which are enzymes

that metabolize citrate and isocitrate, respectively, and decreased levels of succinate and acetyl CoA, components of the TCA cycle. This profile consisting of mostly decreased gene expression related to mitochondrial function, indicated mitochondrial dysfunction in both hypertension and long-lasting hypertension. *Zheleznova et al. (2012)* reported oxygen deficiencies in DS rats using mitochondrial proteomic analyses; mitochondrial protein expression was depressed with decreased oxygen utilization in the renal medullary thick ascending limb of the Dahl rats. Renal mitochondrial damage has also been reported in various models of hypertension, such as the spontaneous hypertension rat, 2-kidney/1-clip model rat, ischemia/reperfusion injury, and renal hypertension in pigs (*Eirin, Lerman & Lerman, 2015*). Therefore, mitochondrial damage may be the hallmark of CKD due to hypertension. In patients with kidney disease, urine metabolomics have revealed that metabolites linked to mitochondria are consistently decreased in the human diabetic kidney (*Sharma et al., 2013*). Others have also reported the downregulation of genes related to oxidative metabolism in patients with CKD on peritoneal dialysis (*Zaza et al., 2013*).

The mechanism underling the renal mitochondrial damage in Dahl rat was not determined in the present study. Hypertension is commonly associated with mechanical stretch, increased production of ROS, and extracellular matrix turnover and fibrosis, which in turn alter the structure and function of cellular organelles including mitochondria. In Dahl rats, enhanced ROS production has been detected in various organs of salt hypertensive Dahl rats including the kidney (*Trolliet, Rudd & Loscalzo, 2001*; *Vokurkova et al., 2015*), heart (*Kato et al., 2010*), liver (*Kato et al., 2012*), and blood vessels (*Swei et al., 1997*; *Trolliet, Rudd & Loscalzo, 2001*). In addition, serum concentrations of pro-inflammatory cytokines are also elevated (*Kato et al., 2012*). These factors may contribute to the development of mitochondrial dysfunction, which in turn causes podocyte injury, tubular epithelial cell damage, and endothelial dysfunction (*Che et al., 2014*). Mitochondrial dysfunction also stimulates TGF-$\beta$ signaling and fibrosis through mitochondrial-derived ROS (*Che et al., 2014*; *Jain et al., 2013*).

There are two clinical implications of the present study. First, its use for biomarker development should be mentioned. Metabolites changed as the disease progressed. Therefore, metabolites in blood and urine samples could be candidates for a biomarker for the early detection of CKD or the prediction of worsening disease (*Cisek et al., 2016*; *Sharma et al., 2013*; *Zaza et al., 2013*). The second implication is the development of therapeutic targets. Though it remains unclear how renal dysfunction cause abnormal metabolic status in mitochondria, the mitochondria could also be a target of therapy for hypertensive kidney disease (*Che et al., 2014*; *Ishimoto & Inagi, 2016*). For example, activation of PGC1$\alpha$ using a transgenic model prevented acute kidney injury (*Tran et al., 2011*). In addition, quinone analogues such as MitoQ are mitochondrial ROS modulators and other antioxidants such as omega-3 polyunsaturated fatty acids, N-acetylcysteine, and allopurinol are all candidates for mitochondrial ROS modulators (*Ishimoto & Inagi, 2016*). The SIRT 1 activator resveratrol improves mitochondrial function and reduces oxidative stress (*Wang et al., 2015*); clinical trials have been planned to investigate usage of these modulators in patients with CKD. The present study as well as previous reports has highlighted possible metabolic therapies for CKD.

This study aimed to characterize one CKD model and the imposed study limitations should be stated. Data on other CKD models were not examined in this study. Metabolome analyses are a snapshot of changes; hence, the flow of metabolites was an estimation. Microarray analysis was not performed, and gene expression was measured by quantitative reverse transcription-polymerase chain reaction (RT-PCR). Therefore, pathway analysis, such as KEGG, could not be performed in the present study. We analyzed the whole kidney including tubules and glomeruli; therefore, we could not isolate the part of the kidney undergoing the metabolic change. In addition, this study was observational and we did not intervene in this model. Therefore, another set of experiments is needed to clarify whether the metabolic changes is the cause or the consequence of kidney injury. However, this model is still useful for preclinical studies since there is an increasing recognition that the kidney plays an important role in complex inter-organ communication, which affects the development of cardiovascular diseases.

## CONCLUSIONS

DS rats fed an HS diet are deemed a suitable model of CKD. Gene expression and metabolites related to energy metabolism and mitochondria in the kidney significantly changed in DS rats with hypertension in accordance with the progression of renal injury.

### Funding

This study was supported by the Japan Society for the Promotion of Science (15K19400). The funders had no role in study design, data collection and analysis, decision to publish, or preparation of the manuscript.

### Grant Disclosures

The following grant information was disclosed by the authors:
Japan Society for the Promotion of Science: 15K19400.

### Competing Interests

The authors declare there are no competing interests.

### Author Contributions

- Yohei Tanada and Junji Okuda conceived and designed the experiments, performed the experiments, analyzed the data, contributed reagents/materials/analysis tools, wrote the paper, prepared figures and/or tables, reviewed drafts of the paper.
- Takao Kato analyzed the data, wrote the paper, reviewed drafts of the paper.
- Eri Minamino-Muta performed the experiments, analyzed the data.
- Ichijiro Murata and Tomoyoshi Soga contributed reagents/materials/analysis tools.
- Tetsuo Shioi conceived and designed the experiments, reviewed drafts of the paper.
- Takeshi Kimura supported the experiment.

## Animal Ethics

The following information was supplied relating to ethical approvals (i.e., approving body and any reference numbers):

Institutional Animal Care and Use Committee of Kyoto University (MedKyo 14183).

## Data Availability

The raw data has been supplied as a Supplementary File.

## Supplemental Information

Supplemental information for this article can be found online at http://dx.doi.org/10.7717/peerj.3352#supplemental-information.

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
