# Peer review of "The metabolic profile of a rat model of chronic kidney disease"

_PeerJ, doi:10.7717/peerj.3352_

## Round 0.1 · original submission · Major Revisions

· Academic Editor

Major Revisions

Thanks for your submission of this paper. Based on the assessment of this article, we propose that you consider the comments from the reviewers, some of which requires consideration of "major" points raised, and resubmit the paper for consideration.

Reviewer 1 ·

Basic reporting

The language of this paper is mostly up to standard for international publication, although I do recommend that the grammar and language be checked. Several sentences were difficult to understand and the reason is that some sentences are simply too long. The sentence starting at line 57 is an example. Another example where editing is required: “...high-salt diet also shows increase of serum…” should rather read: “…high-salt diet also shows an increase of serum…”.

Enough background information with references was provided. However, some adjustment in focus could be favourable for the literature section. The literature review sketches the problem that “the precise molecular mechanisms responsible for renal injury” in CKD is not fully known and that CRS is a symptom of CKD. Hence the model: ? > CKD > CRS. However, their study design and overall focus is more on the reverse, where they induce hypertension to get CKD. The model here is then more like the following: hypertension > CRS > CKD. While it can be expected that the disease (CKD) and symptoms (CRS) progress in parallel (and not in series as I indicated), it might be advantageous to include more discussion and clarification of this model (instead of or in addition to the one mentioned in the problem statement).

Furthermore, some effort in explaining and justifying the animal model was made in the discussion (line 246 – 256) by referring to other animal models. However, the information on the other rodent models is just some loose facts as the authors do not link them or discuss any real differences/advantages.

Lastly, the authors often say that metabolites or gene expression were different between the experimental groups but do not specify whether it is up- or down regulated (E.g. line 223). This means that the reader must constantly consult the figures or tables to understand the paragraph. It would be more informative if the direction of change is also mentioned in the discussion.

Experimental design

The overall design of the investigation is meaningful and falls within the scope of the journal. However, there are some minor (potential) problems or shortcomings that need to be addressed.

First of all, with the study design, it would have been advantageous to use two-factor statistical tests (time vs salt treatment). Thereby, the factors contributing to the metabolite differences could have been sorted and discussed in a more ordered manner. For example, many important metabolites shown in Figure 3, differed between the HS vs LS groups irrespective of time. Since the “treatment effect” is much more important and informative, many of the bar plot figures could have been simplified (by removing the time factor).

It is also unclear if all the kidneys were stored in the same manner (line 101) which could have detrimental effects on the metabolomics results. The authors must clarify this.

Validity of the findings

Line 217: the statement made here is very vague and requires some clarification. In what way do the creatinine and urea levels validate the metabolomics findings? Is it because the hypothesized reduction in respiration results in enhanced protein catabolism?

The sentence starting at line 269 is also vague. Here the authors mentioned that “urine metabolomics revealed that metabolites linked to mitochondria are decreased in the human diabetic kidney”. This conclusion can only be made if the kidneys were analysed.

The rest of the results and discussion is of good quality. The authors also clearly mention the shortcomings and gaps of the current study and clearly state that it have to be clarified.

Additional comments

Figure 3 is one of the punchlines of this investigation but it is not really discussed in much detail. In fact, apart from the conclusion that energy metabolism is affected, the metabolic changes or mechanisms to support this is lacking. Here's an example of how this can be discussed in more detail: reduced respiration leads to a lower NAD:NADH ratio which leads to the accumulation of compound A & B.

By connecting the dots between gene expression and metabolites will make any conclusions more sound.

Reviewer 2 ·

Basic reporting

1. In Fig. 2, it is better to show quantitative results of gene changes.
2. Fig. 3 looks like a conclusive diagram instead of a result summary. Although the raw results of metabolomic analysis uploaded, the primary results need to be summarized appropriately with quantitative data.

Experimental design

1.Metabolomic analysis procedures were insufficiently described in Method. Even it had been described in other references, procedure details for the current study are also obligatory.
2. Primer information should be listed.

Validity of the findings

1. HS model is not a typical model of CKD with CRS. When developing a new model, it is highly recommended that compared it with well-established models in the same lab setting to validate the model. Comparations of marker genes and histology changes can give a portrait of the new model. Furthermore, to verify it is able to recapitulate phenotypes of CKD with CRS, heart functions or histology should be examined in addition to heart rate and SBP (e.g. echocardiography, H&E, IHC). Determinations of compensated or decompensated phase 21w post-HS treatment are also very helpful to characterize the model.
2. Gene changes on RNA level do not guaranty protein level changes. Therefore, protein levels need to be determined for critical genes found by the model.
3. Pathway analysis is highly concerned for the present study. The results can be analyzed in the frame of pathway annotations like KEGG to search potential specific pathways affected.
4. Mitochondrial analysis was partial and not enough to support a major conclusion of the study. Master genes like PGC1a, COX should be summarized and shown in the main figures. Furthermore, to characterize the mitochondrial functional changes better, its morphology, membrane potential and ATP productions are highly concerned.

Additional comments

Abbreviation like CHD should not appear in the title.
Line 237 fed an HS
FigS1B, the acronym of SBP should be annotated in the graph or legend.

---

## Round 0.2 · accepted · Accept

· Academic Editor

Accept

I am happy to inform you that the reviewers are happy with the changes made and that your article is accepted for publication.

Reviewer 1 ·

Basic reporting

The authors improved the overall language of the paper as recommended in the previous review. The quality of some of the figures were also improved.

Experimental design

The authors corrected (or otherwise clarified) some of the issues encountered during the previous review. I believe the design and addition of alternative statistical tests improved the manuscript.

Validity of the findings

The metabolomic results were discussed in more detail and the added information from the two-way statistical tests provided another angle of interpretation that complement the study. Although I still feel that the results could have been discussed more, the shorter discussion balances the length of the paper.

Reviewer 2 ·

Basic reporting

No comment

Experimental design

No comment

Validity of the findings

No comment

Additional comments

All the concerns have been addressed. No more questions.